# Conflict of interest in the peer review process: A survey of peer review reports

**Adham Makarem** [1,2], **Rayan Mroué**[1], **Halima Makarem**[3], **Laura Diab**[1], **Bashar Hassan**[1,4], **Joanne Khabsa**[5], **Elie A. Akl**[6,7] *

1 Faculty of Medicine, American University of Beirut, Beirut, Lebanon, 2 Department of Epidemiology, School of Public Health, Boston University, Boston, Massachusetts, United States of America, 3 Faculty of Arts and Sciences, American University of Beirut, Beirut, Lebanon, 4 Division of Plastic and Reconstructive Surgery, R Adams Cowley Shock Trauma Center, Baltimore, Maryland, United States of America, 5 Clinical Research Institute, American University of Beirut, Beirut, Lebanon, 6 Department of Internal Medicine, American University of Beirut, Beirut, Lebanon, 7 Department of Health Research Methods, Evidence, and Impact (HEI), McMaster University, Hamilton, Canada

* ea32@aub.edu.lb

**Data Availability Statement:** All relevant data are within the paper and its Supporting Information files.

**Funding:** The author(s) received no specific funding for this work.

## Abstract

### Objectives

To assess the extent to which peer reviewers and journals editors address study funding and authors' conflicts of interests (COI). Also, we aimed to assess the extent to which peer reviewers and journals editors reported and commented on their own or each other's COI.

### Study design and methods

We conducted a systematic survey of original studies published in open access peer reviewed journals that publish their peer review reports. Using REDCap, we collected data in duplicate and independently from journals' websites and articles' peer review reports.

### Results

We included a sample of original studies (N = 144) and a second one of randomized clinical trials (N = 115) RCTs. In both samples, and for the majority of studies, reviewers reported absence of COI (70% and 66%), while substantive percentages of reviewers did not report on COI (28% and 30%) and only small percentages reported any COI (2% and 4%). For both samples, none of the editors whose names were publicly posted reported on COI. The percentages of peer reviewers commenting on the study funding, authors' COI, editors' COI, or their own COI ranged between 0 and 2% in either one of the two samples. 25% and 7% of editors respectively in the two samples commented on study funding, while none commented on authors' COI, peer reviewers' COI, or their own COI. The percentages of authors commenting in their response letters on the study funding, peer reviewers' COI, editors' COI, or their own COI ranged between 0 and 3% in either one of the two samples.

**Competing interests:** The authors have declared that no competing interests exist.

## Conclusion

The percentages of peer reviewers and journals editors who addressed study funding and authors' COI and were extremely low. In addition, peer reviewers and journal editors rarely reported their own COI, or commented on their own or on each other's COI.

## Introduction

We recently proposed the following definition of conflict of interest (COI): "A conflict of interest exists when a past, current, or expected interest creates a significant risk of inappropriately influencing an individual's judgment, decision, or action when carrying out a specific duty." [1]. COI may involve a broad spectrum of interests. The financial interests are the most obvious. For example, a researcher may receive significant financial rewards from a pharmaceutical company with interest in the findings of their research. Such financial COIs are common among members of clinical guidelines panels [2, 3]. There is evidence that the quality of the research as well as guidelines may be negatively affected [4].

Non-financial interests can also affect the integrity of research. Such interests include career advancement, fame, social interests, and intellectual beliefs [5]. For instance, an editor may be conflicted when peer reviewing a colleague's work. Intellectual COI is another type of non-financial COI discussed as far as two decades ago [6]. Lately, intellectual COI has been increasingly acknowledged [7] particularly in clinical practice guidelines (CPG) development [8, 9]. It has been defined as "academic activities that create the potential for an attachment to a specific point of view that could unduly affect an individual's judgment about a specific recommendation" [10].

Researchers are expected to avoid and minimize COIs, and disclose them when they exist. The International Committee of Medical Journal Editors (ICMJE) has developed a specific and unified form for disclosure of COIs to facilitate and standardize authors' disclosures. Moreover, there are recommendations on how to manage and declare COIs among authors of clinical practice [11].

Peer reviewers and editors of journals play a key role in assessing and publishing research manuscripts [12]. On one hand, they need to assess the COI of authors and the funding source of the study. On the other hand, journal peer reviewers and editors may have their own conflict of interests that need to be disclosed [13].

In the current era, the standard for scientific publishing is to have research findings evaluated and published through a peer review process [14]. Peer-reviewed biomedical journals are publishing enormous number of articles each year. As of 2012, about 28000 scholarly journals published more than 2 million peer-reviewed articles [15]. An optimal scientific peer review process is essential to maintain the integrity of the scientific research and to support evidence-based practice.

There is increased media attention to the reported conflicts and concerns about the impact of industry-sponsorship [16, 17]. A transparent handling of conflicts of interest is essential for the public trust in the scientific process and the credibility of peer-reviewed published articles [5]. The reporting of authors' disclosure of conflicts of interest in publications has become the standard [13]. In order to facilitate and standardize the process of authors' disclosures, the International Council of Medical Journal Editors (ICMJE) has developed a form for the disclosure of COI [5]. The authors are required to declare via this form all financial and non-financial benefits or personal relationships that might bias their work.

COI issues are relevant to all participants in the peer-reviewed publication process–including peer reviewers, editors, and the editorial board members of the journals. The peer reviewers' role is to critically assess the manuscript, by constructively commenting on the scientific work, and suggesting how to improve it [18]. Moreover, peer-reviewers and editors are expected to reflect and comment on the authors' disclosures of conflicts of interest. There are questions about the effectiveness of the current system of COI disclosures. A randomized controlled study found that providing journal reviewers with authors' conflict of interest information had no significant effect on their rating of the quality of the manuscript [19].

Similarly, journal editors have a core role in managing the review process, through assessing the peer reviewers' reports and making the final decision on acceptance for publication. In addition, editors can significantly impact the integrity, quality and fairness of the peer review process by how they select the peer reviewers and managing any misconducts by authors or reviewers [20]. In fact, an analysis of the Open Payments database found that journal editors commonly accept industry payments which are often large [21]. Some journals have established policies to deal with editors' COI [22]. Editors are also expected to review and consider the authors' disclosures of conflict of interest as part of the peer review process. However, this aspect of their role has not been studied yet.

According to the ICMJE, peer reviewers and editors have to disclose their own conflicts of interest [5]. In some cases, those invited to peer review and editors may need to rescue themselves from being involved. Little is known about the practices and policies of journals regarding disclosures of conflict of interest among peer-reviewers and editors to public [13]. A study assessing the COI policies of health policy and services journals, found only one that described how the COIs of the editorial team are managed during the editorial process [23].

The objective of this study was to assess the extent to which peer reviewers and journal editors address authors' conflicts of interests and study funding. Also, we aimed to assess the extent to which peer reviewers and journals editors reported and commented on their own or each other's COI.

## Methods

We included two samples in this study. First, we included a sample with any type of original research. However, we found that a very low percentage (9%) had at least one author reporting presence of COI. Given this would not allow us achieve the study objective, we decided to collect a second sample restricted to randomized controlled trials (RCTs) as a survey had found that more than half of clinical trials had authors reporting presence of COI [24].

### Eligibility criteria

We included journals that publish in the field of Medicine and in English, are indexed in Medline, and publish all their peer review reports. We included only original articles that are peer reviewed.

### Selecting reports

We used the Directory of Open Access Journals (DOAJ) as an initial list of journals. Then, we filtered the journals by language (English) and subject (Medicine). Then, we selected our samples according to the eligibility criteria. For our first sample, we included the latest two original publications from each journal that had peer review reports. For our second sample, we included the latest two randomized controlled trials (RCTs) that had peer review reports. Teams of two reviewers implemented the selection process in duplicate and independently. They resolved any disagreements through discussion, or with the help of a third reviewer if needed.

## Data abstraction

We collected and managed study data using the Research Electronic Data Capture (REDCap) tool hosted at the American University of Beirut [25]. REDCap is a secure, web-based software platform designed to support data capture for research studies [25]. We also developed detailed instructions. After conducting calibration exercises, teams of two reviewers abstracted data in duplicate and independently. They resolved any disagreements through discussion, or with the help of a third reviewer if needed.

We abstracted the following data:

- Characteristics of the journal (including impact factor, field of study, publisher)

- Characteristics of the publication (primary research vs. systematic review; source of funding)

- Whether the authors reported their COI and the study funding source

- Whether the peer reviewers commented on authors' COI and the study funding source

- Whether the journal editors commented on authors' COI and the study funding source

- Whether the peer reviewers reported their own COI

- Whether the journal editors reported their own COI

- Whether authors and editors commented on the peer reviewers' COI

- Whether authors and peer reviewers commented on the editors' COI

Whenever a COI was reported, we further categorized it according to a framework that we recently published [1]. The framework includes seven types of interests relating to either the individuals (direct financial benefit, benefit through professional status, intellectual, and personal) or their institution (direct financial benefit to the institution, benefit through increasing services provided by the institution, and nonfinancial).

## Data analysis

We exported all data from REDCap to an Excel sheet for data cleaning and consistency checks. We are sharing that Excel sheet to make all raw data available (see S1 Appendix). We conducted quantitative descriptive analyses of all variables. We used percentage for categorical variables and median and interquartile range for continuous variables. We also conducted a thematic analysis of comments extracted.

## Results

### Search results

Fig 1 shows the selection process for the 2 samples included in this study. From the initial list of 150 open peer reviewed journals identified from the DOAJ, we included 72 journals meeting our eligibility criteria.

For our first sample, we included 144 articles, two from each of the 72 eligible journals. For our second sample, 58 journals were eligible and published randomized controlled trials. One of these journals published only one RCT. Consequently, we included a total of 115 articles.

### General characteristics

Table 1 presents the general characteristics of the journals included in the two samples. For both samples, the majority of journal were from the clinical field (80% and 83%), were published in

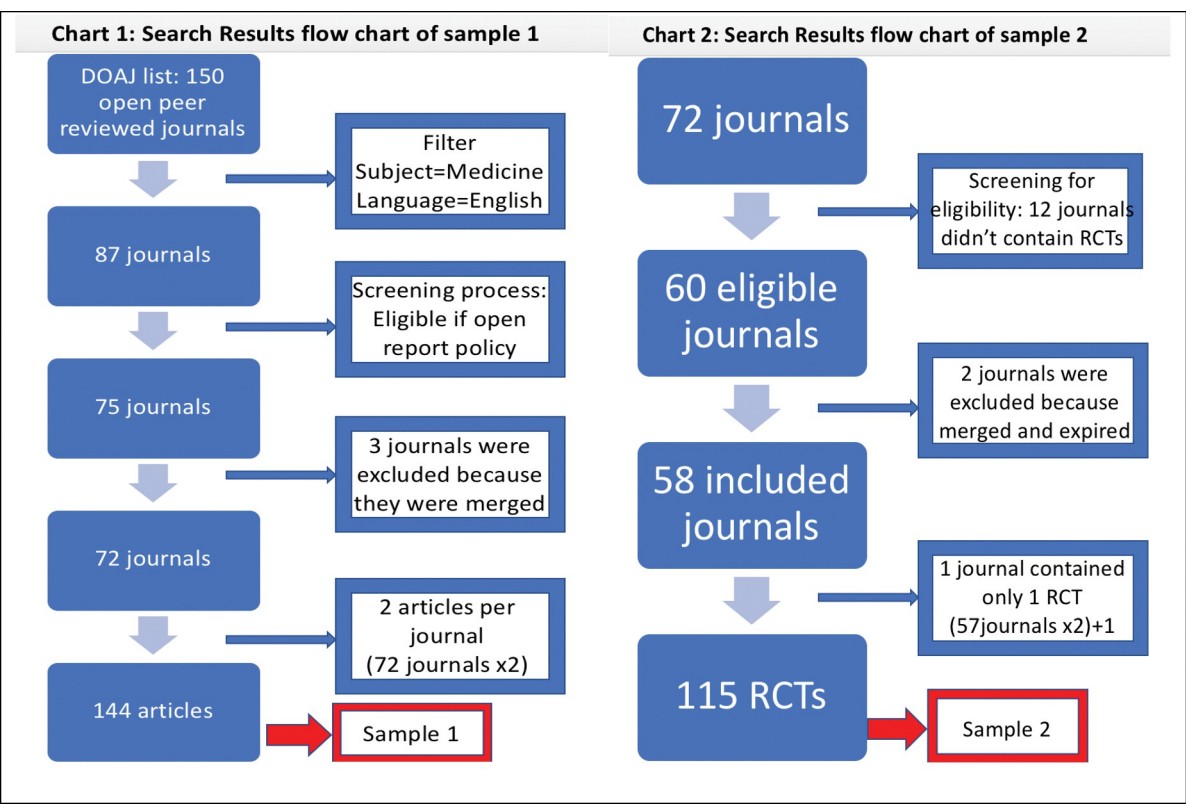

**Fig 1. The selection process for the 2 samples included in this study.**

BioMed Central (BMC) (85% and 89%), and were indexed in Medline (97% and 100%). For the majority of journals peer review was by invitation (99% and 98%), was conducted pre-publication (90% and 91%), and did not reveal the identity of the peer reviewers (56% and 64%).

Table 2 presents the general characteristics of publications included in the two samples. The percentages of RCTs were 3% for sample 1 and 100% for sample 2. For both samples small percentages had COVID-19 as topic (8% and 3%), the median number of authors was 6, the majority had their first author primarily affiliated with academia (87% and 93%), and the majority had the first author affiliated with a high-income country (63% and 61%).

Table 3 presents characteristics of the peer review process. For both samples, the median for number of rounds of peer review, and the number of rounds of revision, and the number of peer reviewers was 2. In the two samples, the median of the number of editors involved was 1. Only 11% and 13% of editors' letters to the authors were posted publicly, and of those 63% and 53% respectively provided the name of the editor.

## Funding of the study

Table 4 presents the characteristics of funding of the included studies. For both samples, the majority of studies were reported as funded (67%and 90%) and included a statement on the funder's role (57% and 59%). The top two sources of funding were governmental (61% and 46%) and internal funding (41% and 46%).

## Declaration of conflict of interest by authors, peer reviewers, and editors

Table 5 summarizes the declaration of conflict of interest by authors, peer reviewers, and editors. For both samples, the majority of studies had all authors report the absence of COI (89%

**Table 1. General characteristics of the included journals.**

| | Sample 1 (N = 72) [1] | Sample 2 (N = 58) [1] |
|---|---|---|
| | n (%) | n (%) |
| Impact factor | Med = 2.674 | Med = 2.652 |
| | IQR = [2.28–3.27] | IQR = [2.28–3.26] |
| Field* | | |
| *Basic sciences* | 34 (47) | 29 (50) |
| *Clinical* | 57 (80) | 48 (83) |
| *Public Health* | 41 (57) | 31 (53) |
| *Health systems and policy* | 23 (32) | 17 (29) |
| *Other* | 4 (5) [2] | 8 (14) [3] |
| Publisher | | |
| *Taylor & Francis* | 4 (6) | 4 (7) |
| *BioMed Central (BMC)* | 61 (85) | 52 (89) |
| *Other* | 7 (9) [4] | 2 (4) [5] |
| Journal indexed in Medline | 70 (97) | 58 (100) |
| Peer review only by invitation process | | |
| *No (open participation)* | 0 (0) | 0 (0) |
| *Yes (invitation needed)* | 71 (99) | 57 (98) |
| *Not reported* | 1 (1) | 1 (2) |
| Peer review report: | | |
| *Pre-publication review* | 65 (90) | 53 (91) |
| *Post-publication review* | 7 (10) | 5 (9) |
| Open identity of peer reviewer | 32 (44) | 21 (36) |

*Total does not add up to 100 due to overlap

[1] sample 1 includes articles of any type of original research. Sample 2 is restricted to only randomized controlled trials (RCTs)

[2] n = 2: Computational and Biotechnologies; n = 1: Biomedical; n = 1: Translational

[3] n = 3: Research Methodologies; n = 2: monitoring and computational modeling; n = 1: Health information technology; n = 1: Health financing and Economics; n = 1: Medical Education and Training

[4] n = 1: Springer; n = 1: Oxford University Press; n = 1: Cambridge University Press; n = 1: BMJ Publishing Group; n = 1: Association for Medical Education in Europe (AMEE); n = 1: Wellcome; n = 1:Physiopedia

[5] n = 1: Springer; n = 1: Wellcome

and 68%). For both samples, the majority of studies had reviewers identified by name (55% and 56%), reported absence of COI (70% and 66%). For both samples, none of the editors whose names were publicly posted reported on COI. Substantive percentages did not report on COI of reviewers (28% and 30%) while only small percentages reported any COI (2% and 4%). Most of these declarations reported individual financial interests with direct financial benefit (see footnotes in Table 5 for more details).

## Commenting on declarations of conflict of interest

Table 6 summarizes the extent of authors, peer reviewers and editors commenting on study funding and conflicts of Interest. The percentages of peer reviewers commenting on the study funding, authors' COI, editors' COI, or their own COI ranged between 0 and 2% in either one of the two samples. 25% and 7% of editors respectively in the two samples commented on study funding, while none commented on authors' COI, peer reviewers' COI, or their own COI. The percentages of authors commenting in their response letters on the study funding,

**Table 2. General characteristics of the included publications.**

| | Sample 1 (N = 144) [6] | Sample 2 (N = 115) [6] |
|---|---|---|
| | n (%) | n (%) |
| Type of research: | | |
| *Randomized controlled trials (RCTs)* | 5 (3) | 115 (100) |
| *Non-randomized studies* | 64 (45) | 0 (0) |
| *qualitative* | 18 (13) | 0 (0) |
| *survey* | 5 (3) | 0 (0) |
| *systematic reviews* | 13 (9) | 0 (0) |
| *case reports* | 17 (12) | 0 (0) |
| *Other*: | 22 (15) | 0 (0) |
| COVID 19 topic | 11 (8) | 2 (3) |
| Number of authors | Med = 6, IQR = [4–8] | Med = 6, IQR = [5–10] |
| First author's primary affiliation: | | |
| *Academic* | 125 (87) | 107 (93) |
| *Governmental* | 8 (6) | 4 (3) |
| *Intergovernmental* | 0 (0) | 0 (0) |
| *Not for profit organization, other than academic* | 6 (4) | 3 (3) |
| *Private for profit* | 5 (3) | 1 (1) |
| *Other* | 0 (0) | 0 (0) |
| Classification of the affiliated country of the first author: | | |
| *High income* | 90 (63) | 70 (61) |
| *Upper-middle income* | 33 (23) | 17 (15) |
| *Lower-middle income* | 15 (10) | 23 (20) |
| *Low income* | 6 (4) | 5 (4) |

[6] sample 1 includes articles of any type of original research. Sample 2 is restricted to only randomized controlled trials (RCTs)

**Table 3. Characteristics of the peer review process.**

| | Sample 1 (N = 144) [7] | Sample 2 (N = 115) [7] |
|---|---|---|
| Number of rounds of peer review | Med = 2, IQR = [1–2] | Med = 2, IQR = [1 –– 2] |
| Number of rounds of revision | Med = 2, IQR = [2–4] | Med = 2, IQR = [2–3] |
| Number of peer reviewers, per study | Med = 2, IQR = [2–2.25] | Med = 2, IQR = [2–2] |
| Number of reviewers, total | 330 | 263 |
| Peer reviewers identified by name | 180 (55) [8] | 146 (56) [8] |
| Number of editors involved, per study | Med = 1, IQR = [1–1] | Med = 1, IQR = [1–1] |
| Public posting of editors' letter to the authors | 16 (11) [9] | 15 (13) [9] |
| Editor identified by name | 10 (63) [10] | 8 (53) [10] |

[7] sample 1 includes articles of any type of original research. Sample 2 is restricted to only randomized controlled trials (RCTs)

[8] denominator is the total number of reviewers: /330 in sample 1 and /263 in sample 2

[9] denominator is the number of editors involved: /144 in sample 1 and /115 in sample 2

[10] denominator is the number of editors who publicly posted letters to the authors: /16 in sample 1 and /15 in sample 2

**Table 4. Funding of the included studies.**

|  | Sample 1 (N = 144) [11] | Sample 2 (N = 115) [11] |
|---|---|---|
|  | n (%) | n (%) |
| Funding status |  |  |
| Not funded | 38 (26) | 9 (8) |
| Funded | 97 (67) | 103 (90) |
| Not reported | 9 (7) | 3 (2) |
| Reported Source(s) of Funding | N = 97 * | N = 103 * |
| *Internally funded* | 40 (41) | 47 (46) |
| *Governmental* | 59 (61) | 47 (46) |
| *Private for Profit* | 11 (11) | 16 (16) |
| *Private not for Profit* | 19 (20) | 22 (21) |
| *Intergovernmental* | 12 (12) | 8 (8) |
| *Academic* | 14 (14) | 8 (8) |
| Statement of the funder's role | 55/97 (57) | 61/103 (59) |

*Total does not add up to 100 due to overlap

[11] sample 1 includes articles of any type of original research. Sample 2 is restricted to only randomized controlled trials (RCTs)

peer reviewers' COI, editors' COI, or their own COI ranged between 0 and 3% in either one of the two samples. Fig 2 below illustrates these results among our 2 samples.

## Discussion

### Summary of findings

In summary, the percentages of peer reviewers and editors commenting on study funding and authors' COI were extremely low. In addition, peer reviewers and journal editors rarely reported their own COI, or commented on their own COI, or on each other's COI.

**Table 5. Declaration of conflict of interest of authors, peer reviewers, and editors.**

|  | Sample 1 [12] | Sample 2 [12] |
|---|---|---|
| Authors reporting of COI | N = 144 | N = 115 |
| *Reports absence of COI* | 129 (89) | 79 (68) |
| *Reports presence of COI (at least 1)* | 13 (9) | 34 (30) |
| *Does not report on COI* | 2 (2) | 2 (2) |
| Peer reviewers' reporting of COI | N = 330 | N = 263 |
| *Reports absence of COI* | 232 (70) | 175 (66) |
| *Reports COI* | 7 (2) [13] | 9 (4) [14] |
| *Does not report on COI* | 91 (28) | 79 (30) |
| *Editor's reporting of COI* | N = 16 | N = 15 |
| *Reports COI* | 0 (0) | 0 (0) |
| *Reports absence of COI* | 0 (0) | 0 (0) |
| *Does not report on COI* | 16 (100) | 15 (100) |

[12] sample 1 includes articles of any type of original research. Sample 2 is restricted to only randomized controlled trials (RCTs)

[13] 4/7 had Individual financial interests with direct financial benefit, 2/7 had Individual personal interests, 1/7 had Individual intellectual interests.

[14] 6/9 had Individual financial interests with direct financial benefit, 3/9 had Individual personal interests.

**Table 6. Commenting on study funding and conflicts of interest by authors, peer reviewers and editors.**

|  | Sample 1 [15] | Sample 2 [15] |
|---|---|---|
| Authors (comments in their response letters) | N = 144 | N = 115 |
| *Authors commenting on study funding* | 5 (3) | 1 (1) |
| *Authors commenting on peer reviewers' COI* | 0 (0) | 0 (0) |
| *Authors commenting on the editors' COI* | 0 (0) [16] | 0 (0) [17] |
| *Authors commenting on their own COI* | 2 (2) | 0 (0) |
| Peer reviewers | N = 330 | N = 263 |
| *Peer reviewer commenting on study funding* | 1 (1) | 4 (2) |
| *Peer reviewer commenting on authors' COI* | 3 (1) | 2 (1) |
| *Peer reviewer commenting on their own COI* | 2 (1) | 1 (1) |
| *Peer reviewer commenting on editors' COI* | 0 (0) | 0 (0) |
| Editors | N = 16 | N = 15 |
| *Editor commenting on study funding* | 4 (25) | 1 (7) |
| *Editor commenting on author's COI* | 0 (0) | 0 (0) |
| *Editor commenting on their own COI* | 0 (0) | 0 (0) |
| *Editor commenting on peer reviewers' COI* | 0 (0) | 0 (0) |

[15] sample 1 includes articles of any type of original research. Sample 2 is restricted to only randomized controlled trials (RCTs)

[16] Denominator is the number of editors who had their letters posted publicly for the authors: /16 in sample 1

[17] Denominator is the number of editors who had their letters posted publicly for the authors: /15 in sample 2

## Strengths and limitations

To our knowledge, this is one of the first studies that examines the extent to which peer reviewers and editors consider study funding and authors' COI, and the extent to which they report on their own COI. To ensure our findings are informative, we went beyond our original plan and included a second sample that involved a higher percentage of articles whose authors

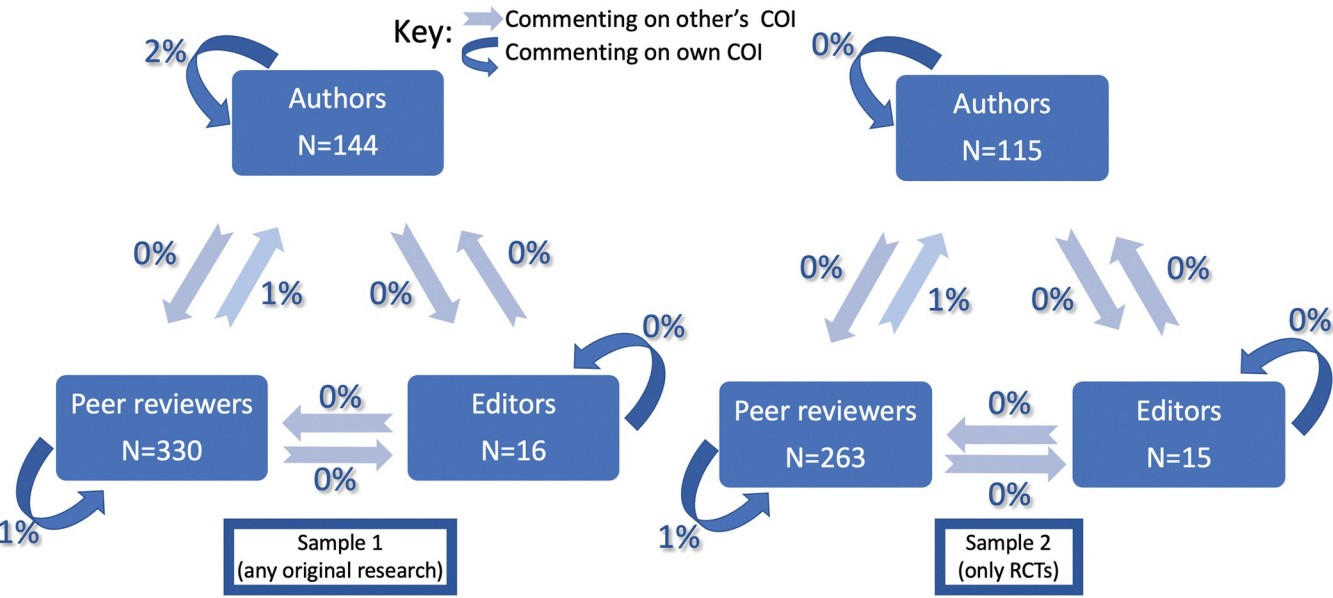

**Fig 2. Commenting on own and other's COI by authors, peer reviewers, and editors.**

reported presence of COI, compared with our first sample. We found consistent results for the two samples. One limitation of this study is that our samples represent the peer review processes of articles that ended up being accepted for publication, but not of those that were not survive the peer review process or those that were published in non-open peer review journal.

## Comparison to similar studies

The percentage of articles with at least one author reporting presence of COI for our first sample was only 9%. This percentage is much lower compared to that of systematic reviews (41%) and clinical trials (57%) [24, 26]. For our second sample, the percentage was higher at 30% because we included only randomized clinical trials. On the second hand, an article by Ralph et al. highlighted the fact that while there are concerns about authors' financial COI in health research, there is little discussion on potential editorial financial COI [27]. In their sample of 20 public health journals, editor and peer reviewer financial COI policies were lacking strict disclosure requirements compared to author financial COI policies [27]. Moreover, another cross-sectional study of high-impact medical specialty journals found that although 99% of those journals required author's COI disclosure, only 12% required editorial's COI disclosure [28]. A review of the COI policies of 72 health policy and services journals found that only one policy described how the COIs of the editorial team are managed during the editorial process [23].

## Implications for practice and research

The results of our study provide insight about the COI disclosure of peer reviewers, authors and editors. One important question is whether the findings are explained by the lack of relevant journal policy or by the non-compliance of editors with such a policy. Reviews of the COI policies of journals in the clinical, surgical and public health areas did not address how the COIs of the editorial team are managed during the editorial process, raising the need for further exploration [29–31].

Our findings may help journals to develop policies to improve how COI is declared and managed during their editorial processes. It would be relevant to conduct qualitative research to explore why some peer reviewers and editors are commenting on authors' COI and on the study funding and others are not. Such study may help in developing strategies to improve COI declaration and management during the peer review process. Also, it would be important to conduct a similar study of articles that underwent the peer review process but did not survive the peer review process or of articles that were published in non-open peer review journal.

## Supporting information

**S1 Appendix.**
(CSV)

## Author Contributions

**Conceptualization:** Adham Makarem, Joanne Khabsa, Elie A. Akl.

**Data curation:** Rayan Mroué, Halima Makarem, Laura Diab, Bashar Hassan.

**Formal analysis:** Adham Makarem.

**Investigation:** Elie A. Akl.

**Methodology:** Adham Makarem, Joanne Khabsa, Elie A. Akl.

**Resources:** Joanne Khabsa, Elie A. Akl.

**Supervision:** Joanne Khabsa, Elie A. Akl.

**Validation:** Adham Makarem, Joanne Khabsa, Elie A. Akl.

**Writing – original draft:** Adham Makarem.

**Writing – review & editing:** Adham Makarem, Rayan Mroué, Halima Makarem, Laura Diab, Bashar Hassan, Elie A. Akl.

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
