## [Decision Letter · Decision Letter 0]

7 Mar 2023

PONE-D-22-27587Conflict of Interest in the Peer Review Process: A Survey of Peer Review ReportsPLOS ONE

Dear Dr. Akl,

Thank you for submitting your manuscript to PLOS ONE. After careful consideration, we feel that it has merit but does not fully meet PLOS ONE’s publication criteria as it currently stands. Therefore, we invite you to submit a revised version of the manuscript that addresses the points raised during the review process.

ACADEMIC EDITOR:This is a very interesting study on an important topic. I commend the authors for their work and urge them to revise the manuscript in light of the comments we received from the reviewers on this. Apologies for the delay in making our decision; we had some difficulties in receiving meaningful reviews from external experts on this submission after having invited a large number of potential reviewers.

We look forward to receiving your revised manuscript.

Kind regards,

Fares Alahdab

Academic Editor

PLOS ONE

Journal Requirements:

2. Please amend your list of authors on the manuscript to ensure that each author is linked to an affiliation. Authors’ affiliations should reflect the institution where the work was done (if authors moved subsequently, you can also list the new affiliation stating “current affiliation:….” as necessary).

Reviewers' comments:

Reviewer's Responses to Questions

**Comments to the Author**

1. Is the manuscript technically sound, and do the data support the conclusions?

Reviewer #1: Yes

Reviewer #2: Yes

Reviewer #3: Yes

Reviewer #4: Yes

2. Has the statistical analysis been performed appropriately and rigorously? 

Reviewer #1: Yes

Reviewer #2: Yes

Reviewer #3: Yes

Reviewer #4: Yes

3. Have the authors made all data underlying the findings in their manuscript fully available?

Reviewer #1: Yes

Reviewer #2: No

Reviewer #3: Yes

Reviewer #4: Yes

4. Is the manuscript presented in an intelligible fashion and written in standard English?

Reviewer #1: Yes

Reviewer #2: Yes

Reviewer #3: Yes

Reviewer #4: Yes

5. Review Comments to the Author

Reviewer #1: The very subject of this paper challenges the peer reviewer to be scrupulous and meticulous in their review. The PLOS peer review process asks me to justify my responses to 4 specific questions.

1. I find the survey approach taken, and the adjustments made, to be appropriate to the question. The authors discuss the limitations of their sample but this is a consequence of issues beyond their control. The conclusions reached are appropriate.

2. The statistical analysis is straightforward, appears to be correctly undertaken and is informative.

3. The authors declare that they have made all the underlying data available, much of it is in the manuscript.

4. The manuscript is clear and easy to follow and written in standard English.

Additional comments

This paper is a salutary reminder of the full extent of our responsibilities as researchers, reviewers and publishers. The research community has come along way in requiring declarations of conflicts of interests from primary researchers, but we clearly have some way to go with how we review this information - saying we have made a declaration should not be enough in itself - but this paper shows us that for the majority of peer reviewers and editors it is - there is not enough checking and alerting the readers of COI going on.

I thought the authors did a good job of presenting their findings in a factual, non-judgmental fashion - they could have said more about the potential implications, but I think they have made the right choice, it is not difficult for the reader to see where these findings lead.

There were some interesting findings that it would have been interesting to hear more about, for example, more on "intellectual COI" and how routinely that is declared. Also, another interesting snippet was that the majority of authors in the papers sampled came from high income countries. Perhaps the acknowledged gap in analysis of papers rejected would tell us more. And finally, across all samples, only one editor used in the peer review process and no COI declared - there are some real lessons for us all in this paper.

Reviewer #2: This is a very important study. Please add an explanation before publication.

Please explain a little bit more about Research Electric Data Capture (REDCap)? I do not understand your description " We developed data extraction form using the Research Electronic Data Capture (REDCap) tool hosted at the American University of Beirut. We also developed detailed instructions. After conducting calibration exercises, teams of two reviewers abstracted data in duplicate and independently. They resolved any disagreements through discussion, or with the help of a third reviewer if needed."

Would you explain the process in detail? Is REDCap validated?

Also, the readers cannot access the system and check the data reproducibility?

Reviewer #3: The manuscript is a technically sound research and the presented data support the conclusions reached. The sampling and sample size are adequate, statistical analyses done are simple but adequately serves the purpose of the research. All the data in the result section were adequately presented. The language of the manuscript is simple and clear, with no grammatical error noted

Reviewer #4: 1. An important theme. Will indeed be relevant towards improving reporting of COI. I have a number of comments seeking clarification and making statements clearer. Please see my comments embedded in the manuscript.

2. It appears that a number of facets of COI reporting in relation to the three key constituencies - authors, reviewers, and editors - and presented in various tables in this manuscript need to be seen in the view of the editorial/journal policies and/or guidelines for authors /submission guidelines. I have tried to indicate such aspects and spots/places in the manuscript as indicative ones in the attached file (the manuscript with my review comments).

3. In this regard, it may mean to present some analysis of the journal policy/submission guidelines/guidelines for authors' of the journals included in this review study. This would help readers to know if many journal simply do not have policy documents that require authors, reviewers and editors to declare their COIs, or if it is to do with the implementation gap, that is, policies are in place but are not implemented with required diligence and rigor at the editorial offices. Suggestion to include such a section. If there might be constraints to present such an analysis, it certainly requires presentation of some insights from the existing scholarship in this area.

3. Under 'What is new?', it says, "This is one of the first studies that examines ... on their own COI." ( p no 4 and later in the submission). However, the basis of such a statement/claim is somewhat unclear as the section '4.3.Comparison to similar studies' compared study findings with some of the existing literature. A more fine tuned an nuance articulation would be more appropriate to keep it close to both the objectives of this review and more importantly findings of the review.

4. It is not clear and not explained as why only two recent papers have selected from each of the journals included in the pool from where the papers are retrieved and included in this review. Please state it explicitly, unless I missed it. As well, it is not adequately as what parameters/criteria employed to determine which two recent papers to be included from many recent papers.

5. Similarly, the rationale for Sample 2, is somewhat weak on two counts: one, the authors might have already undertaken some work to decide on the scope of journals/type of articles to be included in this review for COIs etc; two, it is said that a paper (ref no 25 in the manuscript) led the authors to include RCTs as Sample 2. Both appear to be somewhat ad hoc rationales, especially the latter. Will be helpful and will be needed to present more robust rationale for sample 2. Readers would expect a rationale that is more informed and well-thought which might be the case. However, the current articulation seems like it was 'by chance'.

6. Please you may ignore highlighters in the attached file with my embedded review comments.

6. PLOS authors have the option to publish the peer review history of their article (what does this mean?). If published, this will include your full peer review and any attached files.

Reviewer #1: **Yes: **Dr Safia Qureshi

Reviewer #2: No

Reviewer #3: **Yes: **Oluwatoyin Aduke Babalola

Reviewer #4: **Yes: **Dr Sunita Sheel Bandewar

---

## [Author Response · Author response to Decision Letter 0]

6 Apr 2023

Thank you for your review. We addressed all your comments in our submitted document named: "Response to Reviewers".

---

## [Editor Report · Decision Letter 1]

26 May 2023

Conflict of Interest in the Peer Review Process: A Survey of Peer Review Reports

PONE-D-22-27587R1

Dear Dr. Akl,

I am pleased to see that the issues and areas of concern have been thoroughly and adequately addressed, resulting in an improved manuscript. We’re pleased to inform you that your manuscript has been judged scientifically suitable for publication and will be formally accepted for publication once it meets all outstanding technical requirements.

Kind regards,

Fares Alahdab, MD, MSc

Academic Editor

PLOS ONE

---

## [Editor Report · Acceptance letter]

30 May 2023

PONE-D-22-27587R1 

Conflict of Interest in the Peer Review Process: A Survey of Peer Review Reports 

Dear Dr. Akl:

I'm pleased to inform you that your manuscript has been deemed suitable for publication in PLOS ONE. Congratulations! Your manuscript is now with our production department. 

Kind regards, 

on behalf of

Dr. Fares Alahdab 

Academic Editor

PLOS ONE